

# E-Wybory
## System umożliwiający przeprowadzenie głosowania w wyborach

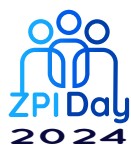

**Autorzy**: Jan Jankowski ⃝ · Michał Starba ⃝ · Krzysztof Saar ⃝ · Krzysztof Wróblewski ⃝

**Opiekun:** Dominika Dudziak-Gajowiak

### Streszczenie

Projekt obejmuje system umożliwiający zdalne głosowanie w wyborach, mający na celu uproszczenie i usprawnienie procesu wyborczego. Oprogramowanie zapewnia bezpieczny sposób oddawania głosów, eliminując konieczność fizycznej obecności wyborców w obwodowych komisjach wyborczych. Użytkownicy mogą jednocześnie uzyskać dostęp do pełnych informacji o kandydatach, procedurach głosowania oraz wynikach wyborczych. System zapewnia również funkcje zarządzania składowymi wyborczymi, takimi jak na przykład: dodawanie kandydatów, komisji, okręgów oraz wyświetlanie statystyk wyborczych. Dodatkowo, projekt gwarantuje bezpieczeństwo danych i integralność procesu głosowania dzięki zastosowaniu mechanizmów uwierzytelniania dwuskładnikowego oraz szyfrowania.

## 1   WSTĘP

Przedsięwzięcie miało na celu rozwiązanie problemu ograniczonego dostępu do procesu wyborczego, wynikającego z konieczności fizycznej obecności wyborców w obwodowych komisjach wyborczych. W kontekście ułatwienia udziału obywateli w wyborach, system ten miał umożliwić zdalne głosowanie, zapewniając jednocześnie bezpieczeństwo danych.

Celem technicznym było stworzenie bezpiecznego i łatwego w użyciu oprogramowania do głosowania zdalnego, które umożliwi wykonywanie czynności wyborczych. Z drugiej strony, celem biznesowym było zwiększenie dostępności procesu wyborczego, umożliwiając szerszemu gronu obywateli udział w wyborach, bez konieczności wychodzenia z domu.

Korzyści z projektu obejmują przyspieszenie procesu głosowania i liczenia głosów, eliminację błędów ludzkich, a także poprawę efektywności i bezpieczeństwa całego procesu wyborczego. Dodatkowo, projekt przyczynił się do zdobycia cennych umiejętności zespołowych, w tym w zakresie pracy w zespole oraz stosowania nowoczesnych rozwiązań technologicznych.

## 2   PRACE ZWIĄZANE Z TEMATEM

### 2.1   Istniejące rozwiązania

Współczesne technologie coraz częściej znajdują zastosowanie w procesach demokratycznych, w tym w systemach głosowania elektronicznego. Jednym z najbardziej rozwiniętych przykładów jest estoński system e-voting, wdrożony już w 2005 roku. Estonia stała się pionierem w umożliwieniu obywatelom głosowania przez internet na masową skalę. System ten oferuje nie tylko transparentność, ale także zaawansowane mechanizmy bezpieczeństwa, takie jak wielokrotne audyty i otwarty kod źródłowy. Dzięki tym rozwiązaniom ponad połowa obywateli w 2023 roku oddała swoje głosy online.

Inne kraje, takie jak Szwajcaria czy Francja, również podejmowały próby wdrożenia e-voting. Szwajcarskie rozwiązania, pomimo innowacyjności i decentralizacji, zostały zawieszone z powodu problemów z bezpieczeństwem i brakiem pełnego zaufania społecznego. Francja natomiast skupiła się na umożliwieniu głosowania obywatelom mieszkającym za granicą, lecz również wstrzymała rozwój projektu z powodu obaw o cyberataki.

Z kolei Rosja korzysta z technologii blockchain w swoich systemach, jednak krytyka dotycząca braku transparentności i ograniczonej dostępności podważa ich wiarygodność. Przykłady te pokazują, że sukces e-voting zależy nie tylko od technologii, ale także od budowania zaufania społecznego i zapewnienia najwyższych standardów bezpieczeństwa.

## 2.2 Wybór technologii

Priorytetem było wykorzystanie technologii zapewniających wydajność, bezpieczeństwo oraz elastyczność systemu. W realizacji projektu zastosowano:

Frontend oraz Backend - Blazor Web App (.NET 8) [4] umożliwia tworzenie zarówno funkcji frontendowych, jak i backendowych w jednym języku programowania (C#). Dzięki temu osiągnięto spójność kodu oraz uproszczono jego rozwój i utrzymanie.

Zarządzanie danymi - MySQL InnoDB Cluster [5] gwarantuje ciągłość działania systemu oraz ochronę danych przed utratą. Rozwiązanie to jest szczególnie istotne w przypadku systemów o krytycznym znaczeniu, takich jak e-voting.

Zabezpieczenia - Wdrożono szyfrowanie SSL/TLS oraz mechanizm JWT (JSON Web Token) do autoryzacji użytkowników. Dodatkowo zastosowano dwuskładnikowe uwierzytelnianie (2FA), co znacząco zwiększa poziom ochrony dostępu do systemu.

Hosting - System został wdrożony na maszynach wirtualnych z systemem Ubuntu LTS udostępnionych przez Politechnikę Wrocławską, co zapewniło stabilność infrastruktury oraz jej łatwe zarządzanie przy użyciu technologii PowerShell.

Pozostałe użyte technologie - Do projektowania graficznego oraz wizualizacji interfejsu użytkownika wykorzystano narzędzia Figma oraz Canva. Do zarządzaniu wersjami kodu oraz organizacji pracy zespołowej użyto Git, Github oraz Jira. Natomiast do stylizacji oraz budowy dynamicznych elementów interfejsu w połączeniu z Blazor Web App zastosowano HTML, CSS i JavaScript.

## 2.3 Ograniczenia czasowe, zasoby i problemy

Podczas pracy nad projektem zespół musiał zmierzyć się z kilkoma istotnymi wyzwaniami. Ograniczenie czasowe było jednym z nich. Projekt realizowano w ramach jednego semestru akademickiego, co narzucało ścisły harmonogram i wymagało starannego planowania. Zespół musiał dokonać trudnych wyborów, rezygnując z mniej kluczowych elementów na rzecz dopracowania najważniejszych funkcji, takich jak bezpieczeństwo i stabilność systemu.

Zasoby, którymi dysponował zespół, były ograniczone, ale dzięki dobrej organizacji pracy udało się je wykorzystać w efektywny sposób. Każdy członek zespołu wniósł swoje umiejętności i wiedzę, co pozwoliło na jasny podział ról i sprawne realizowanie zadań. Współpraca oparta na specjalizacjach poszczególnych osób sprawiła, że zespół mógł skupić się na najważniejszych elementach projektu, skutecznie radząc sobie z napotkanymi trudnościami i ograniczeniami. Istotnym wsparciem okazały się zasoby laboratoryjne udostępnione przez Politechnikę Wrocławską, które posłużyły do wdrażania systemu. Możliwość korzystania z maszyn wirtualnych oraz infrastruktury serwerowej uczelni pozwoliła na bezpieczne testowanie i implementację kluczowych funkcji projektu.

Podczas realizacji napotkano także liczne problemy techniczne i organizacyjne. Wdrażanie zabezpieczeń, takich jak dwuskładnikowe uwierzytelnianie i tokeny JWT, okazało się procesem czasochłonnym, wymagającym precyzyjnej implementacji i licznych iteracji. Dodatkowo brak dostępu do takich narzędzi jak ePUAP czy mObywatel wymusił opracowanie alternatywnych rozwiązań dwuskładnikowego uwierzytelniania użytkowników. Nie obyło się również bez wyzwań związanych z organizacją pracy zespołowej – różne tempo realizacji zadań przez poszczególne osoby wymagało częstych konsultacji i dostosowywania harmonogramu.

## 2.4 Wykorzystane rozwiązania

- Architektura Clean Code [2] z podziałem na warstwy:

  - Warstwa domenowa: encje bazy danych

  - Warstwa aplikacji: DTO

  - Warstwa prezentacji: widoki, viewmodel'e i kontrolery API

  - Warstwa infrastruktury: kontekst bazodanowy

- Architektura Model-View-ViewModel [4]:

  - Dzieląca logikę biznesową od interfejsu użytkownika

- Platforma MS Azure [3]

  - W celu implementacji serwisu obsługującego wiadomości e-mail w aplikacji

# 3 WYNIKI

Przedstawione wyniki pokazują, w jakim stopniu udało się zrealizować założone cele oraz skuteczność zastosowanych rozwiązań technicznych i organizacyjnych.

## 3.1 Zaimplementowane funkcjonalności

- Zarządzanie kandydatami
- Sprawdzenie statusu głosowania wyborcy
- Zarządzanie członkami komisji wyborczych
- Zarządzanie komitetami(partiami) wyborczymi
- Zarządzanie okręgami wyborczymi
- Wyświetlanie szczegółowych wyników wyborczych dla obwodu
- Wyświetlanie i filtrowanie wyników wyborczych
- Zarządzanie obwodami wyborczymi
- Wyświetlanie statystyk głosowania dla obwodu
- Zarządzanie wyborami
- Uwierzytelnianie dwuskładnikowe z wykorzystaniem aplikacji wspierającej TOTP
- Odzyskiwanie hasła
- Wyświetlanie informatora wyborczego
- Logowanie do systemu
- Wylogowanie z systemu
- Rejestracja użytkownika (wyborcy)
- Zmienianie hasła
- Wyświetlanie i filtrowanie statystyk wyborczych
- Funkcjonalność głosowania
- Wysyłanie zaświadczenia potwierdzającego uczestnictwo w wyborach na adres e-mail wyborcy
- Zarządzanie wyborcami
- Generowanie tokenu JWT

## 3.2 Osiągnięte kamienie milowe

- Wykonanie pierwszej wersji aplikacji oraz rozpoczęcie dokumentacji
- Zakończenie głównej implementacji projektu i przejście do testów

## 3.3 Spełnione wymagania biznesowe

- Zdalny dostęp do procesu wyborczego
- Bezpieczeństwo danych i głosów
- Centralizacja informacji wyborczych
- Przyspieszenie i automatyzacja liczenia głosów
- Wsparcie dla organizatorów wyborów

## 3.4   Metryki opisujące projekt

Założenia projektowe nie zakładały wstępnego pokrycia kodu testami, natomiast obecnie prace nad nimi jeszcze trwają. Z uwagi na ograniczony czas trwania pracy nad projektem, priorytetem było zapewnienie jak największej ilości funkcjonalności ujętych w specyfikacji. W ramach weryfikacji poprawności działania aplikacji, skupiono się głównie na testach manualnych poszczególnych komponentów, co pozwoliło na zdiagnozowanie, oraz poprawę błędów w implementacji. Przeprowadzono wstępne testy wydajnościowe aplikacji (po stronie serwerowej) przy użyciu Diagnostic Tools w programie Visual Studio.

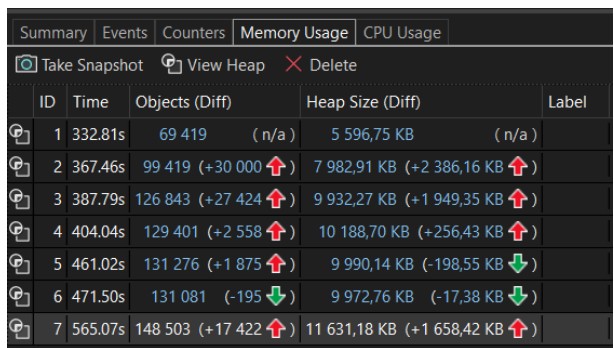

Rysunek 1: Fragment wyników testów wydajnościowych użycia pamięci

## 3.5   Praktyczne zastosowania projektu

Obecnie projekt nie spełnia wszystkich wymagań prawnych, niezbędnych do przeprowadzania wyborów ogólnokrajowych w Polsce. Kluczowym problemem pozostaje brak pełnej gwarancji tajności głosowania [1] w niektórych przypadkach użycia aplikacji. Wdrożenie takiego systemu na poziomie krajowym wymagałoby nie tylko znacznych nakładów finansowych na bezpieczną infrastrukturę, ale także budowy społecznego zaufania, co stanowi ogromne wyzwanie. Aby częściowo rozwiązać te problemy, system został zaprojektowany w sposób umożliwiający kompatybilność z obecnie stosowanymi metodami głosowania w wyborach.

Jednakże, ze względu na specyfikę i ograniczenia, system jest bardziej dostosowany do organizacji wyborów o mniejszej skali, takich jak wybory na burmistrza miasta, wójta, czy sołtysa.

# 4   WNIOSKI

Projekt zakończył się stworzeniem wydajnego systemu głosowania internetowego, który odpowiada na potrzeby zarówno odbiorców technicznych, jak i biznesowych. System oparty na nowoczesnych technologiach, takich jak Blazor Web App [4] i MySQL InnoDB Cluster [5], zapewnia stabilność i łatwość rozwoju, co jest istotne z perspektywy technicznej. Zintegrowane mechanizmy bezpieczeństwa, w tym dwuskładnikowe uwierzytelnianie i szyfrowanie SSL/TLS, zwiększają zaufanie użytkowników oraz chronią dane wrażliwe.

Dla odbiorców biznesowych rozwiązanie oferuje potencjał zastosowania nie tylko w wyborach publicznych, ale także w wewnętrznych procesach decyzyjnych organizacji czy firm. Dzięki intuicyjnemu interfejsowi i wysokim standardom bezpieczeństwa system może usprawniać procesy demokratyczne, jednocześnie minimalizując ryzyko błędów i manipulacji. Wyniki projektu stanowią solidną podstawę do dalszego rozwoju technologii e-voting.

## 4.1   Kierunki rozwoju projektu

· Dostosowanie poziomu zabezpieczeń do nowoczesnych wymogów,

· Obsługa procesów związanych z przeprowadzaniem referendum,

· Integracja z systemami ePUAP i/lub mObywatel, do usprawnienia procesów autentykacji użytkowników

· Zapewnienie zgodności projektu w pełni z prawem w Polsce

· Optymalizacja aplikacji pod kątem wydajności prezentacji treści

## 4.2   Najważniejsze sukcesy projektu

- Ukończenie większości założeń dotyczących projektu określonych na początku semestru

- Zdobyte doświadczenie w zastosowanych technologiach

- Zapoznanie z metodologią Agile w praktyce

- Nabycie umiejętności w pracy zespołowej podczas realizacji projektu informatycznego

- Rozwinięcie umiejętności zarządzania zespołem

# 5   PODZIĘKOWANIA

Chcielibyśmy podziękować:

- Pani mgr inż. Dominice Dudziak-Gajowiak za wsparcie oraz pomoc w organizacji pracy nad projektem

- Panu dr inż. Arkadiuszowi Warzyńskiemu, za udostępnienie zasobów laboratoryjnych na Politechnice Wrocławskiej, służących do wdrożenia systemu

# LITERATURA

[1] *Konstytucja RP art. 96 ust. 2, art. 97 ust. 2 i art. 127 ust. 1.* 1997.

[2] Robert C. Martin. *Clean Code: A Handbook of Agile Software Craftsmanship.* 2008.

[3] Microsoft. *https://learn.microsoft.com/en-us/azure/?product=popular.* 2024.

[4] Microsoft. *https://learn.microsoft.com/en-us/docs/.* 2024.

[5] Oracle. *https://dev.mysql.com/doc/.* 2024.
