# OpenReview forum: "E-Wybory"
_pwr.edu.pl/Wrocław_University_of_Science_and_Technology/2024/ZPI_Day — Wrocław University of Science and Technology 2024 ZPI Day Submission_

### Official Review · Reviewer_dUGb · 2024-12-03
**Abstrakt został starannie przygotowany pod względem strukturalnym i językowym, jednak merytorycznie wymaga poprawy (uściślenia) kilku kluczowych kwestii.**

**Confidence:** 3
**Significance Of Results:** 3
**Overall Quality:** 3

**Compliance With Template:**

4: High Quality – The article contains all the required sections, which are well-written and substantively correct, although minor errors or shortcomings may be present. The overall structure is clear and coherent.

**Description Of Results:**

2: Low Quality – The results are described very superficially and in a general manner. Essential details, usage examples, or evaluations are missing.

**Feedback On Consistency:**

Praca napisana spójnie, choć w wielu miejscach warto było dodać szczegóły techniczne, np. do wymienionych funkcjonalności (są one w losowej kolejności; warto byłoby dodać powiązania tych funkcjonalności z aktorami tworząc klasyczny diagram przypadków).
Przydatny byłby również opis architektury systemu.

**Potential For Development:**

Wymienione w pracy kierunki dalszych prac wydają się naturalną kontynuacją projektu, jednak część z nich ("Dostosowanie poziomu zabezpieczeń do nowoczesnych wymogów" oraz "Zapewnienie zgodności projektu w pełni z prawem w Polsce") wskazuje na dość pobieżne podejście do realizacji tematu - dlaczego te kwestie nie były od początku wzięte pod uwagę (w przesłanej pracy brakuje adekwatnego opisu w tym zakresie)?

**Project Nature Evaluation:**

Projekt wykazuje cechy pracy inżynierskiej. Opisane aspekty pracy wskazują na realizację założonego celu w znacznym stopniu, jednak praktyczne wykorzystanie aplikacji wymaga jeszcze istotnych prac związanych z bezpieczeństwem danych.

**Technical Language Precision:**

4: High Quality – The language is appropriate for a technical report. Terminology is used correctly, and statements are precise, with only minor shortcomings that do not affect the overall clarity.

---

### Official Review · Reviewer_X382 · 2024-12-06
**Recenzja E-Wybory**

**Confidence:** 5
**Significance Of Results:** 4
**Overall Quality:** 4

**Compliance With Template:**

4: High Quality – The article contains all the required sections, which are well-written and substantively correct, although minor errors or shortcomings may be present. The overall structure is clear and coherent.

**Description Of Results:**

4: High Quality – The results are described in detail and supported by usage examples or evaluations. The description is reliable but may lack full depth of analysis.

**Feedback On Consistency:**

Projekt zawiera wszystkie wymagane sekcje.
Powołanie się na istniejące prace mogłoby być opatrzone referencjami, np. do opisu projektów e-governmentowych opisujących wdrożone rozwiązania w przytaczanych krajach.
Informacja o konstrukcji oprogramowania mogłaby zostać rozszerzona, czytelnikowi brakuje modularnego opisu wytworzonych rezultatów, np. w formie diagramu.
Momentami tekst wydaje się być zbyt ustrukturalizowany, sekcje są wyliczeniami elementów bez żadnego komentarza.
Język pracy jest czytelny i nie zawiera elementów żargonowych.

**Potential For Development:**

Projekt ma potencjał na rozwój, W kontekście potencjalnego wdrożenia, które z oczywistych względów w obecnym stanie projektu jest niemożliwe, Autorzy piszą:
"Aby częściowo rozwiązać te problemy, system został zaprojektowany w sposób umożliwiający kompatybilność z obecnie stosowanymi metodami głosowania w wyborach." Warto opisać, jak tę kompatybilność i w jakim zakresie osiągnięto.

**Project Nature Evaluation:**

Zakres prac w projekcie jest ambitny, a sam projekt jest istotny z punktu widzenia potenjanych zastosowań.  wykorzystane techniki, metodyki i narzędzia są uzasadnione właściwe dla przyjętych wymagań.W artykule nie opisano istotnych aspektów związanych z logiką biznesową projetu, np. zagadnienia dot. zapewnienia gwarancji tajności głosowania.

**Technical Language Precision:**

4: High Quality – The language is appropriate for a technical report. Terminology is used correctly, and statements are precise, with only minor shortcomings that do not affect the overall clarity.

---

### Official Review · Reviewer_JHqg · 2024-12-06
**Recenzja projektu e-Wybory**

**Confidence:** 5
**Significance Of Results:** 5
**Overall Quality:** 5

**Compliance With Template:**

5: Very High Quality – The article contains all the required sections, which are written in a very detailed, clear, and error-free manner. The structure is professional and meets expectations, and the content adheres to the highest substantive and formal standards.

**Description Of Results:**

5: Very High Quality – The results are described in detail, clearly and comprehensively, supported by thorough evaluation, analysis, and convincing usage examples. The description meets the highest substantive standards.

**Feedback On Consistency:**

Jakkolwiek mam wiedzę odnośnie całości projektu, oceniając sam abstrakt stwierdzam, że napisany jest w sposób spójny i logiczny. Problem został jasno sprecyzowany. Wyniki zostały przedstawione w sposób właściwy - wskazano zaimplementowane funkcjonalności.

**Potential For Development:**

Jakkolwiek stworzone rozwiązanie wydaje się dedykowane konkretnemu celowi (i jak najbardziej uzasadnione byłoby wdrożenie go), autorzy zauważyli, że potencjał zastosowania nie jest ograniczony stricte do wyborów powszechnych. Wskazano również dalsze kierunki rozwoju, których z przyczyn oczywistych, autorzy zastosować na etapie projektu inżynierskiego (np. integracja z ePUAP i/lub mObywatel).

**Project Nature Evaluation:**

Układ abstraktu jest właściwy, a cel jasno sprecyzowany. Cel został osiągnięty. Narzędzia i technologie są adekwatne. Wysoko oceniam zastosowane rozwiązania w zakresie szeroko rozumianego cyberbezpieczeństwa. Projekt spełnia wymogi projektu inżynierskiego.

**Technical Language Precision:**

5: Very High Quality – The language is entirely appropriate for a technical report. All terms are used correctly and precisely, and the style is professional, clear, and coherent, without any errors or ambiguities.

---

### Decision · Program_Chairs · 2024-12-10

Accept (Poster)